# C2CD4B Evokes Oxidative Stress and Vascular Dysfunction via a PI3K/Akt/PKCα–Signaling Pathway

**DOI:** 10.3390/antiox13010101

**Published:** 2024-01-14

**Authors:** Paola Di Pietro, Angela Carmelita Abate, Valeria Prete, Antonio Damato, Eleonora Venturini, Maria Rosaria Rusciano, Carmine Izzo, Valeria Visco, Michele Ciccarelli, Carmine Vecchione, Albino Carrizzo

**Affiliations:** 1Department of Medicine, Surgery and Dentistry “Scuola Medica Salernitana”, University of Salerno, 84081 Baronissi, Italy; pdipietro@unisa.it (P.D.P.); aabate@unisa.it (A.C.A.); valeria.prete97@gmail.com (V.P.); mrusciano@unisa.it (M.R.R.); carmine.izzo93@gmail.com (C.I.); vvisco@unisa.it (V.V.); mciccarelli@unisa.it (M.C.); cvecchione@unisa.it (C.V.); 2Vascular Physiopathology Unit, IRCCS Neuromed, 86077 Pozzilli, Italy; antonio.damato85@gmail.com (A.D.); eleonora.venturini94@libero.it (E.V.)

**Keywords:** C2CD4B, endothelial dysfunction, oxidative stress, eNOS uncoupling, diabetes

## Abstract

High glucose–induced endothelial dysfunction is an important pathological feature of diabetic vasculopathy. While genome-wide studies have identified an association between type 2 diabetes mellitus (T2DM) and increased expression of a C2 calcium-dependent domain containing 4B (C2CD4B), no study has yet explored the possible direct effect of C2CD4B on vascular function. Vascular reactivity studies were conducted using a pressure myograph, and nitric oxide and oxidative stress were assessed through difluorofluorescein diacetate and dihydroethidium, respectively. We demonstrate that high glucose upregulated both mRNA and protein expression of C2CD4B in mice mesenteric arteries in a time-dependent manner. Notably, the inhibition of C2CD4B expression by genetic knockdown efficiently prevented hyperglycemia–induced oxidative stress, endothelial dysfunction, and loss of nitric oxide (NO) bioavailability. Recombinant C2CD4B evoked endothelial dysfunction of mice mesenteric arteries, an effect associated with increased reactive oxygen species (ROS) and decreased NO production. In isolated human umbilical vein endothelial cells (HUVECs), C2CD4B increased phosphorylation of endothelial nitric oxide synthase (eNOS) at the inhibitory site Thr495 and reduced eNOS dimerization. Pharmacological inhibitors of phosphoinositide 3-kinase (PI3K), Akt, and PKCα effectively attenuated oxidative stress, NO reduction, impairment of endothelial function, and eNOS uncoupling induced by C2CD4B. These data demonstrate, for the first time, that C2CD4B exerts a direct effect on vascular endothelium via a phosphoinositide 3-kinase (PI3K)/Akt/PKCα–signaling pathway, providing a new perspective on C2CD4B as a promising therapeutic target for the prevention of oxidative stress in diabetes–induced endothelial dysfunction.

## 1. Introduction

More than 450 million people worldwide are estimated to be living with diabetes mellitus, a prevalence anticipated to increase by 25% in 2030 and 51% in 2045 [1,2]. The rise in morbidity and mortality among diabetic patients is primarily attributed to the onset of cardiovascular disease (CVD) [3], where vascular complications represent the most severe clinical manifestations of the disease.

Chronic hyperglycemia has been shown to activate oxidative stress–generating machinery, leading to vascular dysfunction in mice [4,5]. Among the vascular cells, endothelial cell (EC) dysfunction emerges as the primary mediator of vascular complications [6]. In diabetic patients, endothelial dysfunction is acknowledged as a critical contributor to vascular disease pathogenesis, often preceding diabetes development [7,8,9,10]. 

Despite these insights, there are few therapies specifically targeting oxidative stress and vascular disease alterations in diabetes. Identifying previously unknown molecules involved in hyperglycemia–induced endothelial damage could pave the way for new therapeutic approaches to prevent or slow down the progression of diabetes–associated vascular complications.

C2CD4B is a protein predominantly found in pancreatic and endothelial cells, where it plays a role in beta cell differentiation [11], and in the regulation of cell architecture and adhesion [12].

Previous studies have suggested that C2CD4B may regulate vascular permeability and factors implicated in thrombotic events [12,13]. A genome-wide significant association was found between the VPS13C/C2CD4A/C2CD4B locus and T2DM risk [14,15,16,17,18]. Subsequent studies revealed that single nucleotide polymorphisms (SNPs) located in close proximity to the C2CD4B gene were associated with diabetes–related traits [19,20], including increased fasting plasma glucose [21], proinsulin levels [22] and impaired beta cell function [23]. 

The present study aims to shed light on the potential effects of C2CD4B on EC and vascular function in mice resistance arteries. Using mice mesenteric arteries, we demonstrate for the first time that recombinant C2CD4B promotes excessive ROS production and impairs nitric oxide signaling, leading to vascular dysfunction through a PI3K/Akt/PKCα–dependent pathway. We show that C2CD4B is required for the development of endothelial dysfunction in hyperglycemic conditions. In terms of its therapeutic implication mechanism, we demonstrate that gene silencing of C2CD4B protects against oxidative stress and endothelial dysfunction induced by hyperglycemic conditions. Thus, targeting C2CD4B opens a new therapeutic avenue for preventing vascular complications in patients with diabetes mellitus.

## 2. Materials and Methods

### 2.1. Reagents

To characterize molecular signaling, mesenteric arteries or HUVECs were pre-treated, or not, before data were obtained with the following: tempol (100 μM for 1 h; Cat. 176141; Sigma-Aldrich, Merck Life Science S.r.l., Milano, Italy); the NOS inhibitor N-ω-nitro-l-arginine methyl ester (L-NAME, 300 μmol/L for 30 min, Sigma-Aldrich, Cat. N5751); the phosphatidylinositol-4,5-bisphosphate 3-kinase inhibitor (wortmannin; Cat. 1232; Tocris Bioscience, Space Import-Export, Milano, Italy); the AKT inhibitor X (Sigma-Aldrich, Cat. 124020); Go6976 (Tocris Bioscience, Cat. 2253). Recombinant human C2CD4B protein (Cat. MBS1165211) was purchased from MyBioSource (S.I.A.L. s.r.l., Rome, Italy).

### 2.2. Cell Culture and Treatment

Human umbilical vein endothelial cells (HUVEC; CRL-4053) were purchased from American Type Culture Collection (ATCC) and cultured in Vascular Cell Basal Medium (ATCC PCS-100-030) supplemented with Endothelial Cell Growth Kit (ATCC PCS-110-041), penicillin (100 U/mL), and streptomycin (100 U/mL). Cells were maintained at 37 °C and 5% CO_2_ in a humidified incubator. To evaluate effects of C2CD4B, cells were cultured for 1 h with the recombinant protein (100 ng/mL). To characterize intracellular signaling, HUVECs were pretreated with the following pharmacological inhibitors: phosphatidylinositol-4,5-bisphosphate 3-kinase inhibitor (wortmannin, 10 μM, 1 h); the AKT inhibitor X (10 μM, 1 h); and Go6976 (500 nM, 30 min).

### 2.3. Mice

All animal experiments were carried out in accordance with the Guide for the Care and Use of Laboratory Animals published by the US National Institutes of Health and approved by the Ethical Committee of Istituto Neurologico Mediterraneo IRCCS Neuromed (Ethical protocol code: 570/2019PR). Mice were maintained in a 22 °C room with a 12 h light/dark cycle, and received food and water ad libitum. Vascular reactivity and molecular studies were performed on 8–10 weeks-old wild-type male C57BL/6 mice (weighing 25–30 g) (Jackson Laboratories, Bar Harbor, ME, USA).

### 2.4. Determination of mRNA Expression Level of C2CD4B

RNA was extracted with TRI reagent (Sigma-Aldrich), quantified by NanoDrop 1000 spectrophotometer (Thermo Fisher Scientific, Milan, Italy), treated with DNase I (Invitrogen, Milan, Italy), and reverse transcribed using a SuperScript VILO Master Mix (Invitrogen, Cat. 11755500). Specific cDNAs were amplified and analyzed using a 7500 Real-Time PCR System (Applied Biosciences, Milan, Italy). Quantification of the relative expression levels was performed using ΔCT calculation. The sequences of primers used for qRT-PCR are the following: C2CD4B, FW: 5′-GACCGCGAAGACAGTGACGAAG-3′; RV: 5′-CGCAGAAGCCGTAAACGGTGC-3′; GAPDH, FW: 5′-CGTCCCGTAGACAAAATGGT-3’; RV: 5′-TCAATGAAGGGGTCGTTGAT-3′.

### 2.5. Vascular Reactivity Studies

Vascular reactivity studies were performed as previously described [24]. In brief, mice mesenteric arteries were mounted in a wire or pressure myograph system containing Krebs solution (pH 7.4 at 37 °C in oxygenated 95% O_2_/5% CO_2_). An amount of 80 mmol/L of KCl was used to evaluate the vasoconstrictive response at the basal level. Phenylephrine (10^−9^ M to 10^−6^ M) was used to reproduce 80% of maximal contraction. Acetylcholine (10^−9^ M to 10^−6^ M) was used to evaluate endothelial–dependent vasodilator response. Mesenteric arteries were treated with recombinant C2CD4B (100 µg/mL) for 1 h. 

### 2.6. Gene Silencing 

Second-order branches of the mesenteric arterial tree were removed from C57BL/6 mice and transfected with siRNA against *C2cd4b* (sc-108918; Santa Cruz Biotechnology Inc.; S.I.A.L. s.r.l., Rome, Italy) and its relative scramble sequence as previously described [25]. Vessels were placed in a Mulvany pressure system filled with Krebs solution and 100 nM of siRNA vector. All vessels were perfused at 100 mmHg for 1 h and then at 60 mmHg for 5 h. Endothelium–dependent and –independent relaxation was assessed by measuring the dilatory responses of mesenteric arteries to cumulative concentrations of acetylcholine (from 10^−9^ M to 10^−5^ M), in vessels precontracted with U46619 at a dose necessary to obtain a similar level of precontraction (80% of initial KCl–evoked contraction) in each vessel. Values are reported as a percentage of lumen diameter change after exposure to the substance.

### 2.7. Evaluation of ROS Production 

To determine eNOS–dependent ROS formation, mesenteric arteries were preincubated with the NOS inhibitor L-NAME (300 μM for 30 min), embedded in Tissue Tek resin, frozen and cryo-sectioned (7 µm) using a cryostat (Leica CM1950, Leica Microsystems, Wetzlar, Germany). In detail, ROS production was evaluated by dihydroethidium staining (DHE, Life Technologies, Carlsbad, CA, USA), as previously described [26]. Briefly, sections were washed in PBS at 37 °C for 10 min and then incubated with DHE (5 µM, Sigma-Aldrich, Cat. D7008) in a humidified chamber protected from light at 37 °C for 30 min. Slices were washed with PBS 1X, and then mounted on a glass slide. Images were observed and acquired under a Nikon Eclipse Ti-E fluorescence microscope (Nikon, Milan, Italy). Fluorescence intensity was measured using ImageJ software.

ROS production in HUVECs was quantified with the membrane-permeable fluorescent probe, Dihydrorhodamine 123 (DHR123) (Thermo Fisher Scientific, Cat. D23806). HUVECs seeded on a microplate were washed with PBS 1X and then treated with DHR123 (10 µM) for 30 min. Fluorescence was determined using an Infinite Pro M200 Tecan microplate reader at a maximum excitation and emission spectra of 507 and 525 nm, respectively. NADPH–mediated superoxide radical production was determined using the lucigenin–enhanced chemiluminescence (ECL) assay, as previously described [25]. Briefly, after reaching 80% confluence, HUVECs were washed with pre-warmed PBS 1X, detached using 0.25% trypsin/EDTA (1 mmol/L), and resuspended in modified HEPES buffer containing 140 mmol/L NaCl; 5 mmol/L KCl; 0.8 mmol/L MgCl_2_; 1.8 mmol/L CaCl_2_; 1 mmol/L Na_2_HPO_4_; 25 mmol/L HEPES; and 1% glucose, pH 7. Subsequently, cells were homogenized, and 100 μg of extract was distributed on a 96-well microplate. Protein content was determined by the Bradford method. The reaction was started by adding NADPH (0.1 mmol/L) and lucigenin (5 μmol/L) to each well. Chemiluminescence was measured using Tecan Infinite Pro M200 microplate reader at 37 °C.

### 2.8. Nitric Oxide Detection 

NO production was evaluated using 4-amino-5-methylamino-2′,7-difluorofluorescein diacetate (DAF-FM, Thermo Fisher Scientific, Cat. D23844). Mesenteric arteries were embedded in Tissue Tek resin, frozen, and cryo-sectioned at 7 µm. Sections were allowed to air-dry at room temperature, fixed with 2% paraformaldehyde for 20 min, and then washed with PBS. Subsequently, they were incubated with DAF-FM (20 µM) for 30 min at 37 °C in a light-protected humidified chamber, washed with PBS, and then mounted on a glass slide. Images were acquired using a fluorescence microscope (Nikon Eclipse Ti-E, Nikon Corp.), and DAF-FM-derived fluorescent intensity was determined using ImageJ software.

### 2.9. Immunoblotting

Protein extracts were separated on SDS-PAGE and then transferred onto a PVDF membrane. The membranes were incubated overnight with the following primary antibodies: anti-eNOS (Cat. #5880; Cell signaling Technology, Danvers, MA, USA); anti-phospho-PI3K (Abclonal, Rome, Italy, Cat. AP0854); anti-phospho-Akt (Cat. sc-7985; Santa Cruz Biotechnology, S.I.A.L. s.r.l., Rome, Italy); anti–β-actin (Cat. ab8226; Abcam, Prodotti Gianni, Milan, Italy); anti–phospho-PKCα (Abcam, Cat. ab76016), anti-α-tubulin (Cat. 627901; Biolegend, Campoverde S.r.l., Milan, Italy); anti-vinculin (Sigma-Aldrich, Cat. V4139); and anti-GAPDH (Santa Cruz Biotechnology, Cat. sc-32233). After a triple wash, membranes were incubated for 1 h with the HRP-conjugated secondary antibodies: anti-rabbit IgG (Invitrogen, Cat. 31463) or anti-mouse IgG (Cat. 31430; Invitrogen, Milan, Italy). Bands were visualized with ECL reagent (Thermo Fisher Scientific, Cat. 32209) according to the manufacturer’s instructions. Immunoblotting data were analyzed using ImageJ software to determine optical density (OD) of the bands. The OD readings of phosphorylated proteins were expressed as a ratio relative to beta-actin or GAPDH.

#### Detection of eNOS Dimer and Monomer

Non-reducing low-temperature SDS-PAGE (LT-PAGE) was performed to detect eNOS dimer and monomer. Briefly, the samples were incubated in O’Farrell’s lysis buffer without (non-reducing) 2-mercaptoethanol at 37 °C for 5 min. Protein extracts were separated on 7.5% SDS-PAGE. Gel and buffers were equilibrated at 4 °C before electrophoresis, and the buffer tank was placed in an ice bath during electrophoresis to maintain the temperature of the gel < 15 °C. Subsequent to LT-PAGE, the gel was transferred, and the immunoblots were performed as previously described [27].

### 2.10. Statistical Analysis

Data are presented as mean ± SEM. Data sets were tested for normality of distribution with the Shapiro–Wilk or Kolmogorov–Smirnov test. Two-sided unpaired Student’s *t*-test was used for comparisons between 2 independent groups. Data groups (comparisons across multiple groups) with normal distributions were compared using one-way ANOVA (with Bonferroni’s correction), unless otherwise indicated. For comparison across different timepoints, data were analyzed using ordinary two-way ANOVA analysis followed by a Bonferroni multiple comparison test. Differences were considered to be statistically significant when *p* < 0.05.

## 3. Results

### 3.1. Genetic Inhibition of C2CD4B Protects against High Glucose–Induced Oxidative Stress and Endothelial Dysfunction

C2CD4B belongs to the C2CD4 family, whose genes are encoded in a susceptibility locus for T2DM [14]. To investigate whether hyperglycemic conditions could modulate C2CD4B expression, we examined mRNA and protein expression in mesenteric arteries treated with normal glucose or high glucose conditions at different time points. RT-qPCR analysis revealed a significant increase in C2CD4B mRNA expression after 3 h of high glucose treatment compared to the normal glucose group, peaking at 6 h post treatment (Figure 1A). While exposure of mesenteric arteries to high glucose for 3 h had no effect on C2CD4B protein expression compared to the normal glucose group, C2CD4B protein was significantly induced after 6 hours of high glucose treatment (Figure 1B).

Interestingly, siRNA–mediated knockdown of C2CD4B prevented the reduction in endothelium–dependent vasodilation induced by 6 h of high glucose in mice mesenteric arteries (Figure 1C). Genetic inhibition of C2CD4B severely blunted increased ROS [dihydroethidium (DHE) cryostaining] and nitric oxide loss [diaminofluorescein-diacetate (DAF-FM)] induced by hyperglycemic conditions in mesenteric arteries (Figure 1D,E). These findings suggest a potential contribution to the development of diabetes-associated vascular complications.

### 3.2. C2CD4B Evokes Endothelial Dysfunction of Mice Resistance Arteries through a ROS–Dependent Mechanism

To eliminate confounding factors associated with hyperglycemic conditions, we conducted ex vivo studies to evaluate whether C2CD4B protein alone could influence endothelial function. Vascular reactivity studies were performed on preconstricted mice mesenteric arteries exposed to the recombinant C2CD4B protein. While 25 and 50 ng/mL did not significantly influence endothelial function, 100 ng/mL induced a significant reduction of acetylcholine–evoked vasorelaxation. A similar result was observed in the presence of 200 ng/mL of C2CD4B. As we noted a prominent impairment in vascular reactivity after incubation with 100 ng/mL of C2CD4B for 1 h (Figure 2A), we decided to use this experimental setting for all subsequent experiments. The experimental setup for assessment of the effects of recombinant C2CD4B is reported in the Appendix A. Notably, this effect was markedly prevented by pretreatment with the antioxidant agent Tempol (Figure 2B), highlighting the crucial role of increased ROS generation in C2CD4B–evoked endothelial dysfunction.

To assess the specific endothelial effects of C2CD4B, we focused our attention on isolated human umbilical vein endothelial cells (HUVECs). C2CD4B significantly increased oxidative stress production after 1 h of exposure (Figure 2C). Additionally, the lucigenin–enhanced chemiluminescence assay clearly indicated the specific involvement of the nicotinamide adenine dinucleotide phosphate (NADPH)–dependent oxidase family in mediating superoxide radical (O_2_^−^) generation in response to C2CD4B treatment (Figure 2D).

### 3.3. C2CD4B Relies on PI3K/AKT Pathway to Induce Endothelial Dysfunction of Mice Mesenteric Arteries

Previous mechanistic studies have shown that under hyperglycemic conditions, EC undergo oxidative stress overproduction and apoptosis through a phosphoinositide 3-kinase (PI3K)/Akt–dependent pathway [28]. Compared to control cells, exposure of HUVECs to 1 h of recombinant C2CD4B significantly increased the expression of phosphorylated forms of PI3K and Akt (Figure 3A,B), indicating the involvement of a PI3K signaling pathway. To clarify this issue, subsequent studies were performed in the presence of the pharmacological inhibitors of PI3K and Akt—wortmannin, and Akt inhibitor X, respectively. Interestingly, both the inhibitors prevented increased NADPH oxidase activation in HUVECs (Figure 3C). At the functional level, wortmannin markedly prevented the impairment of endothelial–dependent vasorelaxation as well as the NO reduction observed in C2CD4B–stimulated mesenteric arteries (Figure 3D,E). This effect was also observed after the pretreatment of vessels with the Akt inhibitor X (Figure 3F), clearly indicating the ability of C2CD4B to drive the activation of the PI3K/Akt signaling cascade, leading to excessive ROS generation and, in turn, to endothelial dysfunction.

### 3.4. Inhibition of Endothelial Nitric Oxide Synthase Prevents C2CD4B–Mediated ROS Generation 

To determine whether the enhanced superoxide formation was dependent on uncoupled eNOS, C2CD4B–treated HUVECs were pre-incubated with the NOS inhibitor, N-ω-nitro-l-arginine methyl ester (L-NAME). Intracellular superoxide production was evaluated using dihydroethidium (DHE) cryostaining in mesenteric arteries. We found that the C2CD4B-induced increase in intracellular superoxide generation was markedly inhibited by L-NAME pre-treatment (Figure 4), suggesting the involvement of eNOS uncoupling in the exaggerated ROS production in response to C2CD4B.

### 3.5. C2CD4B–Mediated eNOS Uncoupling Prevented by PI3K Inhibition

Given the critical role of the dimeric form of eNOS in its functionality [29], we investigated the proportion of the enzyme existing as either dimer or monomer in HUVECs treated with recombinant C2CD4B. Exposure of HUVECs to recombinant C2CD4B for 1 h markedly reduced the dimer/monomer ratio of eNOS compared to control cells, while increasing phosphorylation of eNOS at Thr495, an inhibitory site (Figure 5A,B). Intriguingly, these effects were significantly mitigated by wortmannin pre-treatment (Figure 5A,B; Appendix A), indicating the involvement of PI3K in C2CD4B–mediating eNOS uncoupling.

### 3.6. C2CD4B Induces eNOS Uncoupling and Vascular Dysfunction via a PI3K/Akt/PKCα Signaling Cascade

To further explore the potential role of protein kinase C (PKC), a family of kinases activated in response to high glucose concentration to induce oxidative stress [30]. In particular, we focused our attention on PKCα, known to induce eNOS phosphorylation on its inhibitory site [31]. In addition, PKC upregulation has been reported in metabolic disorders, including diabetes [32].

Recombinant C2CD4B markedly upregulated the protein expression of phosphorylated PKCα, an effect prevented by wortmannin pre-treatment (Figure 6A). As PKCβ is not expressed in HUVECs, Go6976, a specific PKCα pharmacological inhibitor, was employed [33]. Go6976 prevented C2CD4B–induced eNOS dysfunction, as indicated by the preservation of eNOS dimerization (Figure 6B). These findings strongly support the notion that PKCα participates in the C2CD4B cascade, acting downstream of PI3K to mediate uncoupling of eNOS. At the functional level, experiments on mesenteric arteries were conducted, assessing the vascular effects of Go6976 pre-treatment against C2CD4B–induced endothelial dysfunction. Consistent with molecular level observations, Go6976 significantly attenuated endothelial dysfunction, as well as the increased vasoconstriction induced by C2CD4B (Figure 6C,D), clearly indicating the crucial role of PKCα in its harmful effect.

## 4. Discussion

Oxidative stress is a major trigger for diabetes mellitus–induced vascular endothelial dysfunction leading to a reduction in NO bioavailability [34]. There is overwhelming evidence that endothelial dysfunction plays a role in the progression of vascular and end-organ damage in diabetic patients. This makes it a crucial early target for preventing cardiovascular diseases [34]. Diabetes leads to multiple comorbidities, such as heart disease, kidney dysfunction, retinopathy, and impaired wound healing. All these issues can be attributed to vascular disease initiated by the loss of endothelial barrier integrity [35].

Originally described by Warton et al. [12], C2CD4B was, compared to smooth muscle cells and fibroblasts, predominantly identified in human endothelial cells. It was found to be induced after 2 h of treatment with IL-1β.

Genome-wide association studies revealed that single nucleotide polymorphisms (SNPs) in close proximity to the VPS13C, C2CD4A, and C2CD4B genes on chromosome 15q contribute to an increased risk of type 2 diabetes [14,19]. Through human pancreatic islet expression quantitative trait loci (eQTL) analysis, the T2DM–associated risk variant rs7163757 was reported to be linked to increased expression of C2CD4B in alpha cells [15]. However, it remains unknown whether C2CD4B is affected by hyperglycemia conditions and mediates vascular damage in this context.

Here, we show that high glucose markedly increased both mRNA and protein expression of C2CD4B in mice mesenteric arteries in a time-dependent manner. More importantly, the knockdown of C2CD4B protects against high glucose–induced oxidative stress and endothelial dysfunction.

Our findings indicate that C2CD4B is up-regulated by high glucose levels and its inhibition prevents oxidative stress in human endothelial cells and in mice mesenteric arteries.

In diabetic blood vessels, major sources of ROS include the mitochondrial electron transport chain, nicotinamide dinucleotide phosphate (NADPH) oxidase, xanthine oxidase (XO) and uncoupled endothelial nitric oxide synthase (eNOS) [36]. In our study, the increase in ROS generation can, in part, be ascribed to an enhanced expression of NADPH oxidase activity observed in EC treated with the recombinant C2CD4B. Under high glucose conditions, aberrant activation of eNOS, referred to as eNOS uncoupling, produces O^2−^ instead of NO. This superoxide rapidly combines with vascular NO to form peroxynitrite (ONOO−), reducing the bioavailability of NO in vascular EC. This phenomenon results in a vicious cycle that continuously increases superoxide in blood vessels and ultimately impairs endothelium [37]. Importantly, uncoupling of eNOS has also been observed in diabetic patients with endothelial dysfunction [38]. In our experimental setting, an additional contribution to superoxide production came from uncoupled eNOS since pre-treatment with the NOS inhibitor markedly reduced C2CD4B–induced ROS production. Although activation of the PI3K/Akt pathway is critical for maintaining vascular tone and endothelial integrity [39,40], previous studies by Sheu et al. [28] reported that, via a PI3K/Akt–dependent pathway, hyperglycemic conditions may cause ROS generation in HUVECs.

Consistent with these findings, the results of the present study demonstrate that activation of the PI3K/Akt–signaling pathway by C2CD4B can significantly induce oxidative stress in EC and evoke endothelial dysfunction in ex vivo–treated mesenteric arteries. These findings are supported by functional and molecular studies performed on mesenteric arteries, showing that the pharmacological inhibition of PI3K/Akt activity can prevent impaired endothelium–dependent relaxation as well as the reduction of NO levels observed after exposure of vessels to recombinant C2CD4B.

We also observed that C2CD4B exposure in HUVECs resulted in a marked PI3K–dependent reduction in eNOS dimerization and increased phosphorylation of the enzyme on the inhibitory site T495. These are indicative of uncoupled eNOS [41,42]. Moreover, increased eNOS phosphorylation at T495 has been reported to down-regulate NO production [43,44].

Protein kinase C (PKC), an intracellular family of serine/threonine protein kinases, has a crucial role in numerous biological processes, including proliferation, survival, invasion, migration, and apoptosis [45,46]. 

In addition to being associated with several vascular disorders, including hypertension, coronary artery disease, and diabetic vasculopathy [47], activation of PKC has emerged as an important mechanism regulating endothelial dysfunction in diabetes mellitus [48]. In vitro, high glucose concentrations activate PKC and increase superoxide production [49]. Several studies have also demonstrated the ability of PKC to specifically phosphorylate eNOS at T495, inducing oxidative stress and reducing eNOS catalytic activity [31,48]. These concepts are supported by preclinical and clinical studies demonstrating that pharmacological inhibition of PKC ameliorates vascular complications caused by hyperglycemia [50,51].

Our results further show that C2CD4B causes activation of PKCα in HUVEC cells, an effect blunted by PI3K inhibition. We also observed that the PKCα inhibitor Go6976 prevented the decrease in eNOS dimer/monomer ratio, clearly indicating on the one hand, PKCα as a downstream effector of the PI3K/Akt pathway, and an upstream mediator of eNOS uncoupling on the other. These results are consistent with those of previous studies demonstrating the role of PKC activation in mediating eNOS uncoupling and oxidative stress in EC [31]. These data were further corroborated by functional studies, which revealed that pharmacological inhibition of PKCα resulted in significant protection against C2CD4B–induced vascular dysfunction. An exciting observation emerging from our study is that Go6976 did not completely restore endothelial–dependent vasorelaxation, but completely prevented vasoconstriction induced by C2CD4B in mesenteric arteries. These results could be explained by the opposite effects that PKC may elicit on vascular districts. Indeed, PKC is known to influence both vascular relaxation and contraction [52]. For instance, it may mediate NO synthesis but could also induce the release of endothelium–derived constricting factors, thus promoting vasoconstriction [53].

Our data support the notion that C2CD4B is a negative modulator of vascular PKC function and may represent a potential target in vascular disorders. In detail, via PKCα, C2CD4B induces vascular dysfunction through (i) an impairment of endothelium–dependent vasodilation, and (ii) an enhancement of vascular contraction. 

Although newly generated PKC inhibitors have shown promise in the treatment of macular edema [54], retinopathy [55], and microvascular complications [56] in diabetic patients, targeting upstream activators could be a therapeutic strategy for the development of inhibitors able to prevent or delay vascular complications in the early stages of diabetes mellitus.

## 5. Conclusions

Here we present, for the first time, evidence demonstrating that hyperglycemia increased mRNA expression of the diabetic–associated protein C2CD4B. Mechanistically, via activation of the PI3K/Akt/PKCα–pathway, C2CD4B promotes oxidative stress–dependent endothelial dysfunction, driving eNOS uncoupling and NADPH oxidase dysregulation. 

These findings lay the groundwork for further research aimed at deepening our understanding of the molecular mechanism underlying the role of C2CD4B in cardiovascular diseases. Additionally, they suggest the potential therapeutic value of targeting this protein for the prevention of oxidative stress in diabetes mellitus–induced vascular endothelial dysfunction.

## Figures and Tables

**Figure 1 antioxidants-13-00101-f001:**
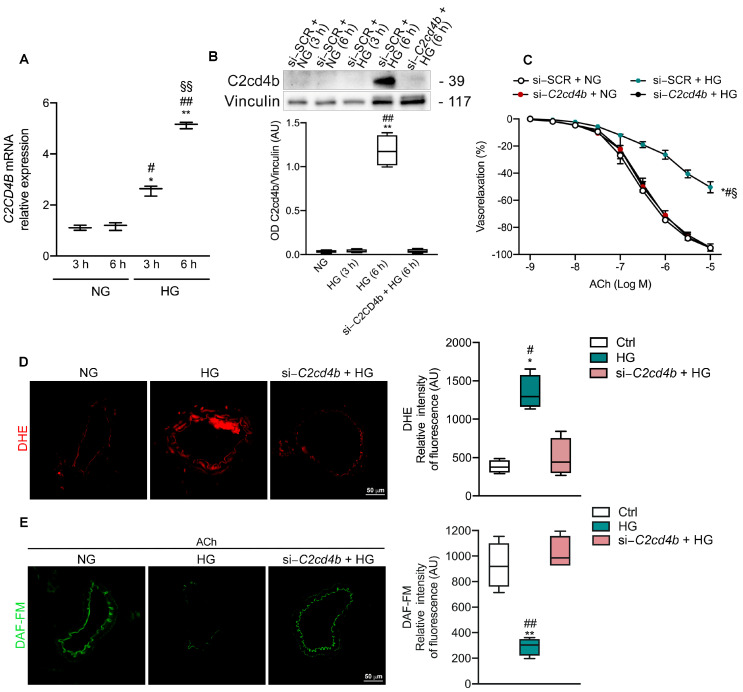
Effect of C2CD4B gene silencing on mice mesenteric arteries exposed to hyperglycemic conditions. (**A**) mRNA expression of C2CD4B determined by quantitative reverse transcription polymerase chain reaction in mice mesenteric arteries treated at different time points (3, 6 h) under normal glucose (NG, 5 mM) or high glucose conditions (HG, 30 mM); (*n* = 3). (**B**) Representative western blot and densitometric analyses evaluating protein levels of C2CD4B in mice mesenteric arteries treated with normal glucose (NG) or high glucose (HG) at different time points (3, 6 h), pretransfected with either scramble siRNA (NG 3 h; NG 6 h; HG 3 h and HG 6 h), or specific siRNA against C2CD4B (Si−C2CD4B) before HG 6 h; (*n* = 4). Non-parametric Kruskal–Wallis test with Dunn’s correction was used. (**C**) Acetylcholine (ACh)–evoked vasorelaxation in mice mesenteric arteries exposed to NG or HG conditions and pretransfected with siRNA against C2CD4B (Si–C2CD4B) or scrambled siRNA (Si–SCR); (*n* = 3). (**D**) ROS levels were detected by DHE in mice mesenteric arteries treated with normal glucose (NG), high glucose (HG) alone, or pretransfected with siRNA against C2CD4B (Si–C2CD4B). Box plots show relative fluorescence intensity, (AU, arbitrary units); (*n* = 4). (**E**) NO levels were detected by DAF-FM in mice mesenteric arteries treated with normal glucose (NG), high glucose (HG) alone, or pretransfected with siRNA against C2CD4B (Si–C2CD4B). Box plots show relative fluorescence intensity, (AU, arbitrary units); (*n* = 4). Unless otherwise stated, statistical analyses were performed using one-way or two-way ANOVA followed by Bonferroni post-hoc test. * *p* < 0.05; ** *p* < 0.01; # *p* < 0.05; ## *p* < 0.01; § *p* < 0.05; §§ *p* < 0.01.

**Figure 2 antioxidants-13-00101-f002:**
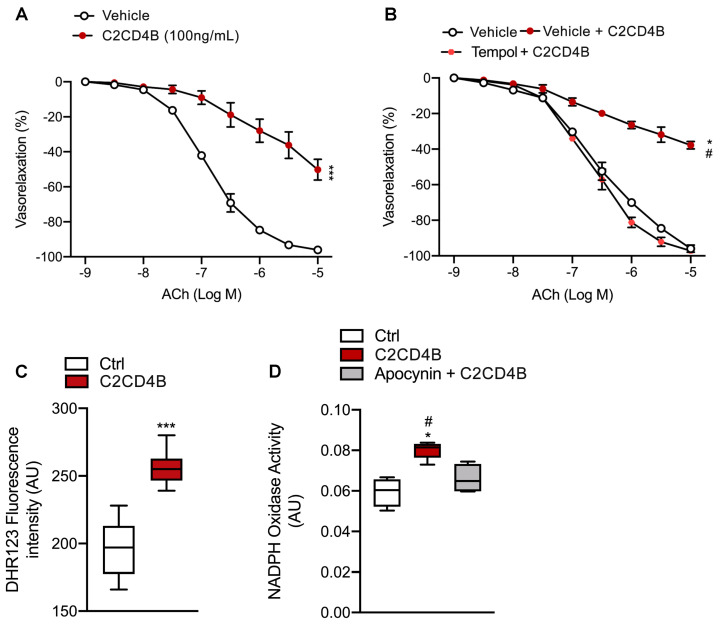
Effect of recombinant C2CD4B on mice mesenteric arteries and human EC. (**A**) Acetylcholine (ACh)–evoked vasorelaxation in mice mesenteric arteries exposed to vehicle or recombinant C2CD4B; (*n* = 3). (**B**) Acetylcholine (ACh)–evoked vasorelaxation in mice mesenteric arteries exposed to vehicle or recombinant C2CD4B in presence or absence of the antioxidant agent tempol; (*n* = 3). (**C**) Cellular ROS levels assessed by determining dihydrorhodamine 123 (DHR) fluorescence intensity (AU, arbitrary units); (*n* = 4–6). (**D**) NADPH–induced lucigenin chemiluminescence (data are expressed as increase in chemiluminescence per minute in arbitrary units) in HUVECs treated with 100 ng/mL of C2CD4B for 1 h; (*n* = 4–6). Statistical analyses were performed using Student’s *t*-test, one-way or two-way ANOVA followed by Bonferroni post-hoc test. * *p* < 0.05; *** *p* < 0.001; # *p* < 0.05.

**Figure 3 antioxidants-13-00101-f003:**
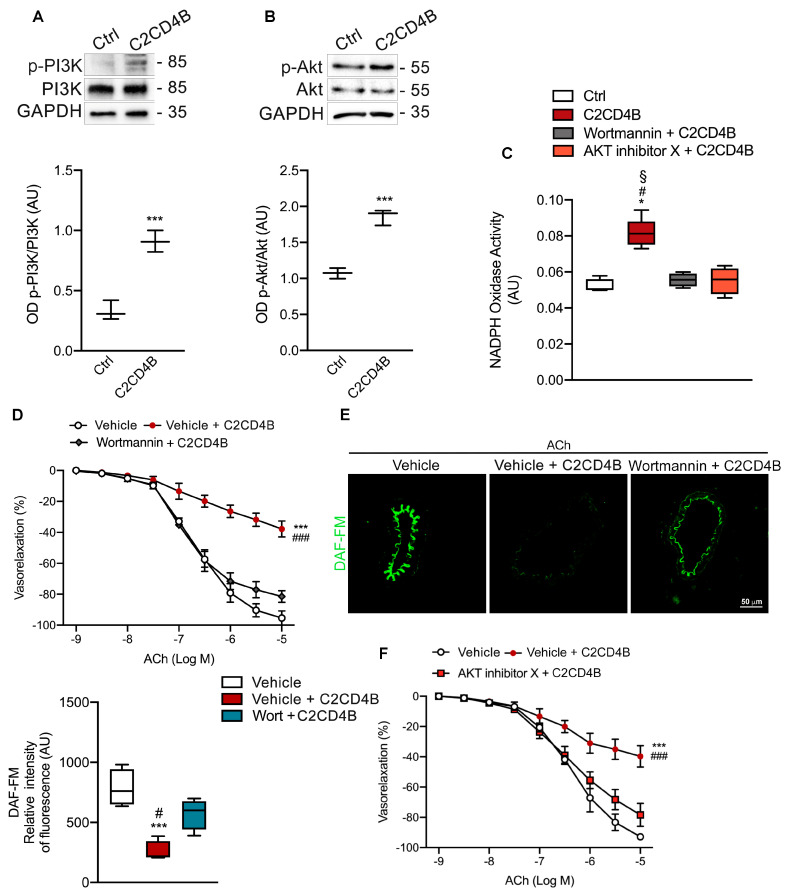
C2CD4B promotes endothelial dysfunction of mice mesenteric arteries through PI3K/Akt signaling pathway. (**A,B**) Representative western blot and densitometric analyses of three independent experiments evaluating protein levels of (**A**) phospho–PI3K and (**B**) phospho–Akt protein expression in HUVECs treated with vehicle (ctrl), or recombinant C2CD4B. (**C**) NADPH–induced lucigenin chemiluminescence (data are expressed as increase in chemiluminescence per minute in arbitrary units) in HUVECs treated with 100 ng/mL of C2CD4B in the presence or absence of wortmannin, or AKT inhibitor X; (*n* = 4–5). (**D**) Acetylcholine (ACh)–evoked vasorelaxation in mice mesenteric arteries exposed to vehicle or recombinant C2CD4B in the presence or absence of wortmannin (*n* = 3). (**E**) NO detection by DAF-FM in mice mesenteric arteries treated with vehicle, C2CD4B alone, or pre-treated with wortmannin. Box plot shows relative fluorescence intensity; (*n* = 3). (**F**) ACh–evoked vasorelaxation in mice mesenteric arteries exposed to vehicle or recombinant C2CD4B in the presence or absence of AKT inhibitor X; (*n* = 3). Statistical analyses were performed using Student’s *t*-test, one-way or two-way ANOVA followed by Bonferroni post-hoc test. * *p* < 0.05; *** *p* < 0.001; # *p* < 0.05; § *p* < 0.05; ### *p* < 0.001.

**Figure 4 antioxidants-13-00101-f004:**
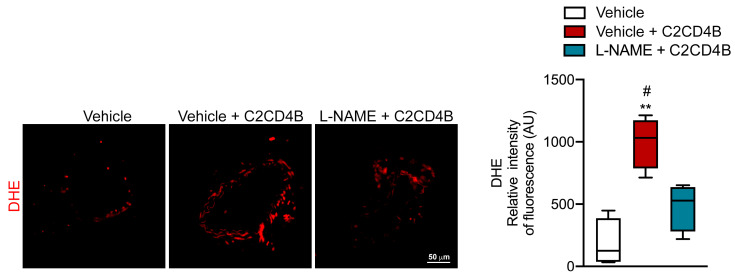
C2CD4B induces eNOS uncoupling in mesenteric arteries. ROS production detected by DHE in mice mesenteric arteries treated with vehicle, C2CD4B alone, or pre-treated with L-NAME. Box plot shows relative fluorescence intensity; (*n* = 4). Statistical analyses were performed using one-way ANOVA followed by Bonferroni post-hoc test. ** *p* < 0.01, # *p* < 0.05.

**Figure 5 antioxidants-13-00101-f005:**
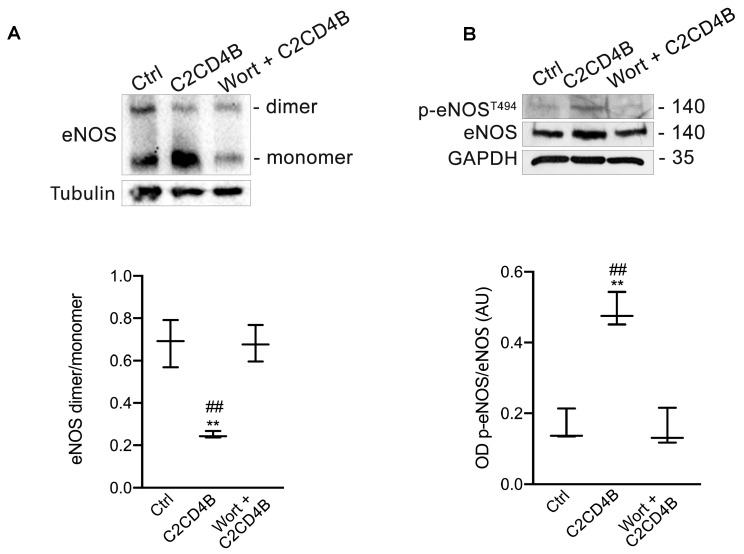
Wortmannin prevents C2CD4B–induced eNOS uncoupling in EC. (**A**) Representative western blot and densitometric analyses of three independent experiments evaluating eNOS dimer (280 kDa) and eNOS monomer (140 kDa) protein expression in HUVECs treated with vehicle (ctrl), recombinant C2CD4B alone, or pre-treated with wortmannin. Dimer/monomer eNOS were examined by non-reducing low-temperature SDS-PAGE. (**B**) Representative western blot and densitometric analyses of three independent experiments evaluating protein levels of phosphorylated eNOS at Thr 495 (p-eNOS^T495^), and total eNOS expression in HUVECs treated with vehicle (ctrl), recombinant C2CD4B alone, or pre-treated with wortmannin. Statistical analyses were performed using one-way ANOVA followed by Bonferroni post-hoc test. ** *p* < 0.001; ## *p* < 0.001.

**Figure 6 antioxidants-13-00101-f006:**
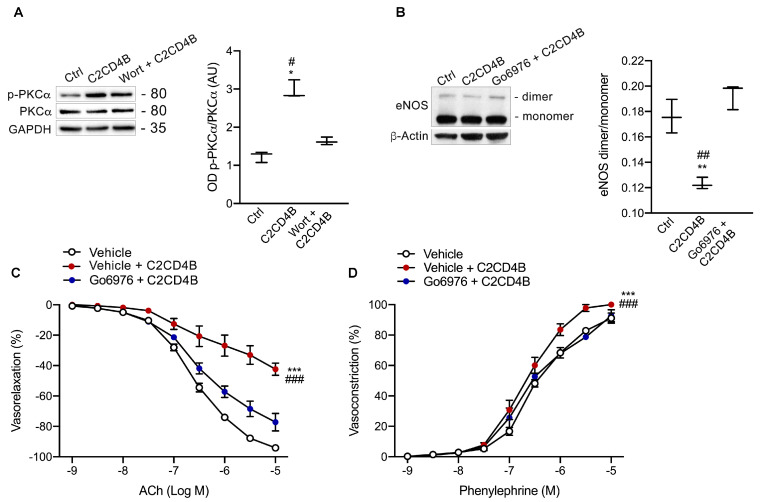
Pharmacological inhibition of PKCα reverses eNOS uncoupling and prevents vascular dysfunction induced by C2CD4B. (**A**) Representative western blot and densitometric analyses of 3 independent experiments evaluating the expression of phosphorylated form PKCα in HUVECs treated with vehicle (ctrl), recombinant C2CD4B alone, or pre-treated with wortmannin. (**B**) Representative western blot and densitometric analyses of three independent experiments evaluating eNOS dimer (280 kDa) and eNOS monomer (140 kDa) protein expression in HUVECs treated with vehicle (ctrl), recombinant C2CD4B alone, or pre-treated with the pharmacological inhibitor of PKCα, Go6976. Dimer/monomer eNOS were examined by non-reducing low-temperature SDS-PAGE. (**C**) Acetylcholine (ACh)–evoked vasorelaxation in mice mesenteric arteries exposed to vehicle or recombinant C2CD4B in the presence or absence of the PKCα inhibitor, Go6976; (*n* = 3–4). (**D**) Dose–response curves to phenylephrine of mice mesenteric arteries exposed to vehicle or recombinant C2CD4B in the presence or absence of Go6976; (*n* = 3–4). Statistical analyses were performed using one-way or two-way ANOVA followed by Bonferroni post-hoc test. * *p* < 0.05; ** *p* < 0.01, *** *p* < 0.001; # *p* < 0.05; ## *p* < 0.01; ### *p* < 0.001.

## Data Availability

The data presented in this study are available on request from the corresponding author.

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
