# Peer review of "C2CD4B Evokes Oxidative Stress and Vascular Dysfunction via a PI3K/Akt/PKCα–Signaling Pathway"

_antioxidants, 2024, doi:10.3390/antiox13010101_

Round 1

Reviewer 1 Report

Comments and Suggestions for Authors

The authors present here an interesting study on the effects of C2CD4b on vascular dysfunction. The study is generally well done, although there are a few issues that must be addressed before it is publishable.

No detail is provided on the recombinant C2CD4b, which forms a vital part of the experiments. Is this human C2CD4b or mouse? Is it purchased commercially (in which case details are needed) or produced in house (in which case either details of its preparation and purification are needed, or citation of a previous work in which this is described).

It is crucial that authors provide details of how many times each experiment is repeated (n=?) in the figure legends,

The authors use ImageJ to quantify Western blotting, which is OK provided that blots are not overexposed. When this is the case, there is no longer a linear relationship between protein level and signal. This is a problem in figure 6C, as well as maybe 5a.

In figure 3, the authors demonstrate a difference in PI3K phosphorylation, but is there also a different in total PI3K expression?

The authors need to clarify if the mice used are male, female or a combination in the Methods section

Minor points

There is a lot of variation in whether the authors say C2cd4bm C2CD4b or c2cd4b.  I suggest using C2CD4b for human protein, and C2cd4b for mouse.

Sometime the authors refer to T494 phosphorylation and sometimes T495. Are these different events? The same events in human and mouse proteins? Or just a mistake?

The discussion is mostly a description of the results, and would benefit from more description of how these results fit into the existing literature.

Author Response

REVIEWER #1

The authors present here an interesting study on the effects of C2CD4b on vascular dysfunction. The study is generally well done, although there are a few issues that must be addressed before it is publishable.

  • No detail is provided on the recombinant C2CD4b, which forms a vital part of the experiments. Is this human C2CD4b or mouse? Is it purchased commercially (in which case details are needed) or produced in house (in which case either details of its preparation and purification are needed, or citation of a previous work in which this is described).

Response 1: We apologize to this Reviewer for our inaccuracy. In the revised version of the methods section, we have defined that we used a commercially available recombinant human protein.

  • It is crucial that authors provide details of how many times each experiment is repeated (n=?) in the figure legends,

Response 2: As suggested by the Reviewer, we have defined the n number for each experiment in the Figure legends.

  • The authors use ImageJ to quantify Western blotting, which is OK provided that blots are not overexposed. When this is the case, there is no longer a linear relationship between protein level and signal. This is a problem in figure 6C, as well as maybe 5a.

Response 3: We agree with the comment raised by the reviewer. As it is known, the gel conducted to visualize the eNOS monomer dimer is performed in low temperatures under non-reducing conditions; this procedure does not always allow clear visualization of the eNOS dimer signal. In the revised version of the manuscript, we repeated the experiment in Figure 5A that was over-exposed. Our results confirmed the previously obtained finding, in which treatment with C2CD4B induces a reduction in eNOS dimerization, whereas treatment with wortmannin in the presence of C2CD4B is able to promote the recovery of dimerization. Moreover, quantification of western blotting was conducted by using Image Lab. The same software was also used for the quantization of Figure 6B, which confirms that the inhibition of PKCa promotes the recovery of eNOS dimerization under C2CD4B treatment.

  • In figure 3, the authors demonstrate a difference in PI3K phosphorylation, but is there also a different in total PI3K expression?

Response 4: We really appreciate the criticism of this Reviewer. In the new version of the manuscript, we have performed new experiments analyzing PI3K expression and its phosphorylated form in response to treatment with recombinant C2CD4B. These new analyses reveal the increase of phosphorylated form of PI3K following incubation with recombinant C2CD4B, whereas there was no difference in total PI3K expression levels among the two groups. Accordingly, these data have been added in the revised results section of the manuscript (Figure 3A).

  • The authors need to clarify if the mice used are male, female or a combination in the Methods section

Response 5: We apologize to the Reviewer for our inaccuracy. This point has been clarified in the new version of the manuscript. In detail, we have reported: “only vessels from male mice were used for the study. The use of female mice would have resulted in the use of a much higher number of animals due to the impact of sex steroid hormones on vascular function. In compliance with the principle of the Reduction, we decided to conduct the study only on male mice. The analysis of gender differences in the action of C2CD4B on vasculature may represent a topic for a follow-up study.

Minor points

  • There is a lot of variation in whether the authors say C2cd4bm C2CD4b or c2cd4b. I suggest using C2CD4b for human protein, and C2cd4b for mouse.
  • gene symbols are reported italicized, first letter upper case all the rest lower case for

Response: We apologize to this Reviewer for the inaccuracy. We used “C2CD4B” referring to the recombinant protein. For mus musculus gene symbol and protein symbol, we used the first letter in upper case and all the rest in lower case; instead, for the homo sapiens gene symbol and protein symbol, we used all letters in upper case. Gene symbols are reported as italicized.

  • Sometime the authors refer to T494 phosphorylation and sometimes T495. Are these different events? The same events in human and mouse proteins? Or just a mistake?

Response: We apologize to this Reviewer for our mistake. The Threonine 494 residue of mouse endothelial nitric oxide synthase corresponds to Threonine 495 of human endothelial nitric oxide synthase. Accordingly, we have thoroughly checked the whole manuscript and corrected the mistake.

  • The discussion is mostly a description of the results, and would benefit from more description of how these results fit into the existing literature.

Response: We thank this Reviewer for his/her criticism. We have applied the suggested modification accordingly. We hope that these revisions could meet all the concerns.

Reviewer 2 Report

Comments and Suggestions for Authors

In this paper the authors presented high glucose treatment for 6 hours increased C2ch4b expression in in wild-type mouse mesenteric arteries and in HUVECs. This caused increased superoxide anion production and in turn impairs endothelium-dependent relaxations. Blockade of C2ch4b by gene silencing normalized these alterations.  The study presents novel information. However, there are several major concerns that must be clarified further by the authors:

1) Important information regarding C2ch4b is missing. There are no data showing that the selected incubation time (1 hour) and concentration (100 µg/mL) of recombinant protein C2ch4b is optimal in your experimental set-up. The authors need to include the optimization data for concentration and incubation time in the manuscript.

2) Where did the authors purchase recombinant protein C2ch4b (company name, catalog number)?  This information must be included in the manuscript.

3) For gene silencing studies the authors need show the proof that 6 hours treatment with C2ch4b-siRNA efficiently deleted protein expression of C2ch4b. Western blot study should have been done. This information must be included in the manuscript.

4) Before using statistical analyses, the authors didn't perform normality test for such small sample sizes. If the data don't pass normality test, then they must be analyzed by non-parametric tests.

5) The authors used bar graphs for presentation of results in the figures. Bar graphs are applicable for frequencies in categorical variables only but not for presenting continuous/quantitative measures. Therefore, the results need to be presented in box plots or dot plots instead of bar graphs. (Weissgerber TL et al. PLoS Biol. 2015 Apr 22;13(4):e1002128).

6) The number n of experiments is missing the legends of Figures 1-6.

7) Molecular mechanisms such as Western blot studies were performed only in HUVECS.  It is unclear why the authors did not study protein expressions in mouse mesenteric arteries?

Author Response

REVIEWER #2

In this paper the authors presented high glucose treatment for 6 hours increased C2ch4b expression in in wild-type mouse mesenteric arteries and in HUVECs. This caused increased superoxide anion production and in turn impairs endothelium-dependent relaxations. Blockade of C2ch4b by gene silencing normalized these alterations.  The study presents novel information. However, there are several major concerns that must be clarified further by the authors:

1) Important information regarding C2ch4b is missing. There are no data showing that the selected incubation time (1 hour) and concentration (100 µg/mL) of recombinant protein C2ch4b is optimal in your experimental set-up. The authors need to include the optimization data for concentration and incubation time in the manuscript.

Response 1: We thank the reviewer for this observation. To investigate the dose-dependent effects of C2CD4B recombinant protein, we performed vascular reactivity studies using increasing concentrations of C2CD4B (25, 50, 100, and 200 ng/mL) for 1 hour prior to perform dose-response curves to acetylcholine-evoked vasorelaxation.  While 25 and 50 ng/mL did not significantly influence endothelial function (Supplementary Figure 1), 100 ng/mL induced a significant reduction of acetylcholine-evoked vasorelaxation. A similar result was observed in the presence of 200 ng/mL of C2CD4B. As we noted a prominent impairment in vascular reactivity after preincubation with 100 ng/mL for 1 hour, we decided to use this experimental setting for all subsequent experiments.

2) Where did the authors purchase recombinant protein C2ch4b (company name, catalog number)?  This information must be included in the manuscript.

Response 2: We apologize to this Reviewer for our inaccuracy. In the revised version of the methods section, we have specified that we used a commercially available recombinant human protein.

3) For gene silencing studies the authors need show the proof that 6 hours treatment with C2ch4b-siRNA efficiently deleted protein expression of C2ch4b. Western blot study should have been done. This information must be included in the manuscript.

Response 3: We thank the Reviewer for his/her suggestion. Accordingly, the efficiency of siRNA-mediated knockdown of C2cd4b was verified by western blot studies and reported in the revised version of the manuscript (Figure 1B). The results obtained demonstrate the efficiency of gene silencing in deleting C2cd4b protein expression after 6 hours of high glucose incubation.

4) Before using statistical analyses, the authors didn't perform normality test for such small sample sizes. If the data don't pass normality test, then they must be analyzed by non-parametric tests.

Response 4: We apologize with this Reviewer for the insufficient description of the analytical approach. In accordance to his/her suggestions, in the revised version of the manuscript, we have better described the statistical analysis performed. The Shapiro-Wilk or Kolmogorov-Smirnov tests were used to evaluate the normality distribution of investigated parameters.

5) The authors used bar graphs for presentation of results in the figures. Bar graphs are applicable for frequencies in categorical variables only but not for presenting continuous/quantitative measures. Therefore, the results need to be presented in box plots or dot plots instead of bar graphs. (Weissgerber TL et al. PLoS Biol. 2015 Apr 22;13(4):e1002128).

Response 5: We thank the Reviewer for his/her valuable suggestions. Accordingly, all the results have been presented in box plots in this revised version of the manuscript.

6) The number n of experiments is missing the legends of Figures 1-6.

Response 6: As correctly suggested by the Reviewer, we have added the n number for each experiment in the Figure legends.

7) Molecular mechanisms such as Western blot studies were performed only in HUVECS.  It is unclear why the authors did not study protein expressions in mouse mesenteric arteries?

Response 7: We appreciate the comment of the Reviewer. It is well known that acetylcholine is a gold standard neurotransmitter used in vascular reactivity studies to evaluate endothelial mechanotransduction. In particular, its interaction with the M3ACh receptor (M3AChR) at the endothelial level acts to trigger calcium release from the internal store in endothelial cells, thus promoting nitric oxide production, and in turn, artery relaxation. Based on our results demonstrating an impairment of endothelial-dependent vasorelaxation of mice mesenteric arteries exposed to recombinant C2CD4B protein, we employed human umbilical vein endothelial cells as our model system to investigate the molecular mechanisms recruited by C2CD4B.

In addition, in line with the principles of the "3Rs" of animal research, it is important to emphasize that mice mesenteric artery is about 180 mm in diameter and 2 mm in length (the size required for mounting the vessel on the pressure myograph). Due to its small size, it is impossible to obtain useful material to conduct a molecular analysis from a single sample, but it is necessary to make a mesenteric pool in order to obtain sufficient protein quantities for a useful sample. Using a pool of 4 mesenteric arteries for each treatment, western blot analysis revealed a significant increase of C2cd4b protein in mesenteric arteries exposed to 6 hours of HG as compared to normal glucose condition (NG, 5mM) and its silencing in the presence of specific siRNA (Figure 1B).

We hope that the revised version of the manuscript will now be acceptable for publication by this Reviewer.

Reviewer 3 Report

Comments and Suggestions for Authors

The authors explore the impact of elevated glucose levels on endothelial dysfunction, a crucial component of diabetic vasculopathy. The study specifically investigates the involvement of C2CD4B in this process. This hypothesis is substantiated by previous research establishing a connection between type 2 diabetes mellitus and elevated C2CD4B expression. The authors first note that increased glucose levels result in the upregulation of C2CD4B mRNA expression in the mesenteric arteries of mice in a time-dependent manner. Furthermore, they suggest that silencing C2CD4B effectively prevents hyperglycemia-induced oxidative stress, endothelial dysfunction, and diminished nitric oxide (NO) bioavailability. In contrast, the administration of recombinant C2CD4B induces arterial endothelial dysfunction, elevated reactive oxygen species (ROS), and reduced NO production. The authors propose that C2CD4B directly influences vascular endothelium through a PI3K/Akt/PKC-signaling pathway.

One primary issue pertains to the English language. I propose a revision and improvement of English language to enhance the readability of the manuscript. Notably, authors should clearly describe each figure and panel in both the text and the legend of the figures. This is often overlooked, making the interpretation of the data arduous. Overall, the manuscript exhibits numerous experimental and conceptual pitfalls, necessitating thorough revision and improvements. The study itself could be important in supporting the role of C2CD4B in vascular dysfunction. Unfortunately, in its current form, the manuscript fails to convey this message smoothly. Listed below are some concerns and comments regarding the manuscript.

Major

·         Figure 1. In the gene silencing experiments, the authors state that they used C2CD4B-specific siRNAs; however, at the same time, they mention the use of a control ('scrambled') vector (line 109). It is not clear to the reader whether these experiments were conducted through vector-based siRNA transfection or simply using dsRNA molecules. In addition, there is a need for a clear demonstration of the efficacy of RNAi experiments on the expression of the C2CD4B protein.

·         Moreover. Does the increasing level of C2CD4B mRNA correlate with an increased level of the protein? The authors should include this data.

·         Figure 1. The various panels of this figure 1 require more detailed descriptions. It is unacceptable that there is almost no explanation of the results shown in the individual panels. The authors cannot simply state that '...siRNA-mediated knockdown of C2CD4B prevented endothelial dysfunction, oxidative stress, and nitric oxide loss under high glucose conditions (Figure 1B-1D).” There is a need for a more detailed description of the results.

·         Figure 1B. The information conveyed by the data in Figure 1B is not clear. Based on the graph, there seem to be no differences between the C2CD4B silencing samples and the scrambled controls, both in the presence of high glucose and under normal glucose levels. Please provide an explanation.

·         Figure 2. Legend of panels A and B should say "Vehicle+C2cd4b", "Tempol+C2cd4b", while "basal" should be "Vehicle". Overall, the legend for Figure 2 appears completely mislabeled. Panels "E and F" do not exist in the figure but are reported in the legend. This causes confusion.

·         Figure 3. Please note that the experiments conducted to evaluate PI3K and pAKT expression are not described in the text. Could you provide information on the treatment times and the culture media used for these treatments?

·         Figure 5. Since the structural uncoupling of eNOS is a crucial feature in endothelial dysfunction, the effect of C2cd4b on eNOS monomers and dimers should be more convincing. Additionally, I would expect a lower amount of oligomers in the absence of β-ME (the authors state that they do not add β-ME). The authors can test the effect of increasing the concentration of β-ME on the ratio between monomer and dimer. I would expect that an increasing concentration of β-ME will inhibit the formation of oligomers. Therefore, the author can test protein samples reconstituted in sample buffer with or without β-Mein the absence or in the presence of Wortmaninn.

·         Have authors investigated the effect of C2cd4b on the expression of NADPH subtypes?

Minor

·         Line 46. “Despite these insights, there are few therapies specifically target oxidative stress vas-46 cular disease alterations in diabetes.”. It should be “Despite these insights, there are few therapies specifically targeting oxidative stress and vascular disease alterations in diabetes.”

·         Line 52. Since the abbreviation “EC” was introduced for the first time in the introduction (at line 42), you can use it instead of 'endothelial cells" through the text. Please check other abbreviations as well.

·         c2cd4b, C2CD4b, and C2CD4B are used through the text. Please, always use the same formatting.

·         There is no indication of how the HUVECs were cultured. Please, add appropriate section to methods.

·         Figure 3 shows some panels that are not clearly described in the text or in the legend.

·         Figure 5. (Panel A) It would be useful to reiterate that this data was generated by running low-temperature SDS-PAGE

·         Line 167. The statement "…the samples were incubated in O'Farrell lysis buffer without 2-mercaptoethanol at 37°C for 5 min" raises a question. Indeed, to my knowledge, the original recipe of O'Farrell lysis buffer contains 2-mercaptoethanol. If this is true, it is not correct to state that the samples were incubated in the absence of 2-mercaptoethanol. If so, please revise the section.

Comments on the Quality of English Language

Round 2

Reviewer 1 Report

Comments and Suggestions for Authors

The authors have done an admirable job of addressing my comments, and I consider the paper ready for publication.

Author Response

We thank this Reviewer for his/her valuable and insightful comments, which have been very helpful in improving the manuscript.

Reviewer 2 Report

Comments and Suggestions for Authors

The authors have addressed all of the reviewer's concerns and the manuscript is significantly improved. I do not have any further comments. 

Author Response

We very thank this Reviewer for the constructive suggestions and we are delighted to read your comments about our manuscript.

Reviewer 3 Report

Comments and Suggestions for Authors

I appreciate the authors for revising their manuscript in accordance with the reviewer's comments. Nevertheless, there are still a few points that require attention.

·         Additional comment related to Authors’response 2. Please note that the control (scramble sequence) is absent in the figure. As mentioned earlier, the experiment is intended to demonstrate the impact of high glucose treatment on both control and siRNA-transfected cells. To clarify, the western blot panel should incorporate high glucose treatment in both scramble siRNA-transfected cells and specific siRNA-transfected cells.

·         Additional comment related to Authors’response Response 3. Once again, the experiment lacks control. If the author is referring to Figure 1B, what was stated above still applies.

·         Additional comment related to Authors’response Response 4. Please note that in the revised figure legend for Figure 1, the last part of the legend (letter B) is mislabeled and appears unrelated to the indicated panel in the figure.

·         Additional comment related to Authors’response Response 8. These data provide more information. I recommend incorporating them into the revised version of the manuscript.

Comments on the Quality of English Language

Minor editing of English language required

Author Response

#REVIEWER 3

I appreciate the authors for revising their manuscript in accordance with the reviewer's comments. Nevertheless, there are still a few points that require attention.

1. Additional comment related to Authors’response 2. Please note that the control (scramble sequence) is absent in the figure. As mentioned earlier, the experiment is intended to demonstrate the impact of high glucose treatment on both control and siRNA-transfected cells. To clarify, the western blot panel should incorporate high glucose treatment in both scramble siRNA-transfected cells and specific siRNA-transfected cells.

Response 1. We apologize for our inaccuracy. In the revised version of the paper, we have clearly specified that protein levels of C2cd4b were evaluated in mice mesenteric arteries under normal or high glucose (NG or HG, respectively) conditions and pretransfected with either scramble siRNA (NG 3hrs; NG 6 hrs; HG 3 hrs and HG 6 hrs) or with specific siRNA against C2cd4b (Si-C2cd4b) before HG 6 hrs; (n = 4). We have now added this information into the revised figure 1 and its related figure legend.

2. Additional comment related to Authors’response Response 3. Once again, the experiment lacks control. If the author is referring to Figure 1B, what was stated above still applies.

Response 2. We thank the reviewer for pointing out our mistake. Accordingly, in the revised version of the manuscript we have added the missing specific control.

We hope that this revised version could meet the reviewer's requirements.

3. Additional comment related to Authors’response Response 4. Please note that in the revised figure legend for Figure 1, the last part of the legend (letter B) is mislabeled and appears unrelated to the indicated panel in the figure.

Response 3. We apologize for the lack of clarity in the mentioned figure legend. We have revised it accordingly.

4. Additional comment related to Authors’response Response 8. These data provide more information. I recommend incorporating them into the revised version of the manuscript.

Response 4. We are very grateful to the Reviewer. Accordingly, these data have been added in the revised supplementary section of the manuscript.